# Comparison of the Prognostic Value of Inflammatory and Nutritional Indices in Nonmetastatic Renal Cell Carcinoma

**DOI:** 10.3390/biomedicines11020533

**Published:** 2023-02-12

**Authors:** Tomoyuki Makino, Kouji Izumi, Hiroaki Iwamoto, Suguru Kadomoto, Yoshifumi Kadono, Atsushi Mizokami

**Affiliations:** Department of Integrative Cancer Therapy and Urology, Kanazawa University Graduate School of Medical Science, 13-1 Takara-machi, Kanazawa 920-8640, Japan

**Keywords:** renal cell carcinoma, inflammation, nutrition, neutrophil-to-lymphocyte ratio, platelet-to-lymphocyte ratio, C-reactive protein-to-albumin ratio, prognostic nutritional index, geriatric nutritional risk index, survival, metastasis

## Abstract

Several markers that reflect inflammation and nutritional status have been associated with oncological outcomes in many tumors. This study aimed to describe the impact of pretreatment inflammatory and nutritional indices on the oncological outcomes in nonmetastatic renal cell carcinoma (RCC). A total of 213 Japanese patients with nonmetastatic RCC at Kanazawa University Hospital between October 2007 and December 2018 were included. The inflammatory and nutritional indices, including neutrophil-to-lymphocyte ratio (NLR), platelet-to-lymphocyte ratio (PLR), C-reactive protein-to-albumin ratio (CAR), prognostic nutritional index (PNI), and geriatric nutritional risk index (GNRI), were retrospectively analyzed. The optimal cutoffs for NLR, PLR, CAR, PNI, and GNRI were 2.18, 153.7, 0.025, 48.4, and 98, respectively. According to Kaplan–Meier curves, elevated NLR, PLR, CAR, and GNRI correlated with increased metastasis, while NLR and PNI correlated with worse overall survival (OS). In multivariate analysis, high CAR was an independent poor risk factor for metastasis (hazard ratio (HR), 3.08; 95% confidence interval (CI), 1.24–7.67; *p* = 0.016). Furthermore, high NLR showed an independent prognostic factor for worse OS (HR, 3.96; 95% CI, 1.01–15.59; *p* = 0.049). The pretreatment inflammatory and nutritional indices such as NLR and CAR might be promising prognostic factors for nonmetastatic RCC.

## 1. Introduction

The prognosis of renal cell carcinoma (RCC) is influenced by anatomical, histological, clinical, and molecular factors. For localized RCC, several risk scores and nomograms can be used, including the Stage, Size, Grade, and Necrosis Score (SSIGN) [1], a modified version of the SSIGN score (Leibovich score) [2], the University of California Los Angeles Integrated Staging System [3], and the Karakiewicz nomogram [4]. Cancer prognosis is influenced not only by the behavior of the tumor but also by the patient’s physical condition, for which inflammatory and nutritional status are particularly important. 

Systemic inflammation plays an important role in almost every stage of tumor development, progression, and metastasis [5]. Thus, the systemic inflammatory response is significantly associated with the outcomes of patients, and related inflammatory indices, such as the neutrophil-to-lymphocyte ratio (NLR), platelet-to-lymphocyte ratio (PLR), and C-reactive protein (CRP)-to-albumin ratio (CAR), may be used as biomarkers to effectively estimate the prognosis of patients with cancer [6,7]. Furthermore, accumulating evidence has shown that nutritional and immune status, represented by the prognostic nutritional index (PNI), serum albumin, CRP, and body mass index (BMI), play a role in the development and progression of malignant tumors, which in turn affects survival [8]. Thus, there is no doubt that the presence of systemic inflammation and nutritional status are associated with various cancer outcomes. These prognostic biomarkers, including the NLR, PLR, CAR, PNI, and geriatric nutritional risk index (GNRI), have been identified as independent predictors of survival in patients with RCC [7,9,10,11,12,13,14,15]. This study aimed to retrospectively examine the prognostic value of pretreatment NLR, PLR, CAR, PNI, and GNRI for survival in patients with nonmetastatic RCC after curative surgical treatment in a Japanese population.

## 2. Patients and Methods

### 2.1. Patients

Patients with nonmetastatic RCC (T1–T4, N0, M0) at Kanazawa University Hospital between October 2007 and December 2018 were included in this retrospective study. The inclusion criteria were set as follows: (I) age of 18 years or older; (II) confirmed imaging or histologic diagnosis of RCC; (III) complete electronic medical records including clinical laboratory tests within one month before surgery. We excluded patients who did not undergo surgical therapy.

This study was approved by the Medical Ethics Committee of Kanazawa University (2018-116). All research was performed in accordance with relevant guidelines and regulations and the Declaration of Helsinki. The requirement for informed consent was waived by the Medical Ethics Committee of Kanazawa University due to the observational nature of the study using only existing data. Instead, information about this study will be posted on the Kanazawa University Hospital website, and patients are free to revoke their consent at any point.

### 2.2. Data Collection and Variable Definitions

Baseline clinical data at the time of surgical treatment, including age, sex, BMI, NLR, PLR, CAR, PNI, and GNRI, were collected. Additionally, baseline oncological data, including pathological tumor stage and size, histological subtype, histological nuclear grade, and micro/lymphovascular invasion, were obtained. Pathological stage evaluation was performed according to the tumor–node–metastasis classification of malignant tumors (eighth edition) by the Union for International Cancer Control (2017).

NLR and PLR were obtained preoperatively by calculating the relationship between the absolute number of neutrophils, platelets, and lymphocytes. CAR was calculated by dividing the serum CRP level by the serum albumin level.

PNI was defined based on the serum albumin level and lymphocyte count using the following formula: 10 × serum albumin (g/dL) + 0.005 × total lymphocyte count of peripheral blood (per mm^3^). GNRI was calculated as follows: [14.89 × serum albumin (g/dL)] + [41.7 × (body weight/ideal body weight)] [16]. The ideal body weight (kg) was calculated using the Lorentz equation: ideal body weight = height (cm) – 100 − [(height − 150)/4] for men and height (cm) – 100 − [(height − 150)/2.5] for women, and weight/ideal body weight = 1 when the weight exceeds the ideal body weight [16].

Overall survival (OS) was defined as the time from the date of surgical treatment to the date of death from any cause. Metastasis-free survival (MFS) was measured as the time from the date of surgical treatment to the date of the first detection of metastasis from RCC.

### 2.3. Assessment of the Inflammatory Biomarkers and Nutrition Status

Reproducible markers of systemic inflammation include the NLR, PLR, and CAR. Although the optimal cutoff points for NLR, PLR, and CAR vary in previous studies, the optimal cutoff values in this study were determined using the point closest to (0,1) on the receiver operating curve (ROC) [17]. The nutrition-related risk was assessed by PNI and GNRI. Although the cutoff values of PNI were not uniform in several studies, ranging from 38.5 to 51.62 [18], this study set the cutoff values according to ROC analysis. GNRI was used to systematically assess the risk of malnutrition. The GNRI formula above defined four categories: high risk (<82), moderate risk (82 to <92), low risk (92 to ≤98), and no risk of malnutrition (>98) [16]. In this study, malnutrition in patients with RCC was defined as GNRI ≤98.

### 2.4. Statistical Analysis

The differences between the patients’ clinicopathological characteristics and inflammatory and nutritional indices were compared using the Mann–Whitney test. OS and MFS were estimated using the Kaplan–Meier method and compared using the log-rank test. Univariate and multivariate analyses were performed using Cox proportional hazards models to evaluate the association of NLR, PLR, CAR, PNI, and GNRI with MFS and OS. Statistical analyses were performed using GraphPad Prism version 6.07 (GraphPad Software Inc., San Diego, CA, USA) and IBM SPSS Statistics version 25 (IBM Corp., Armonk, NY, USA). A *p*-value <0.05 indicated statistical significance.

## 3. Results

### 3.1. Patient Background

Data on 213 patients with T1–T4, N0, M0 RCC were extracted. The median follow-up period for the study population was 4.45 years (range, 0.03–13.11 years). Of the 213 patients, 155 (72.8%) were male, and the median age was 63 years (range, 18–85 years). Most patients (199, 93.4%) underwent total or partial nephrectomy, and 14 (6.6%) underwent ablative therapies, such as cryoablation and radiofrequency ablation. No patients in this study received postoperative adjuvant therapies. Of the 213 patients, 172 (80.8%) had clear cell RCC, and 33 (15.5%) had other histologic types.

### 3.2. The Optimal Thresholds for NLR, PLR, CAR and PNI

Using OS as the end point for NLR, PLR, CAR, and PNI, the optimal cutoff values were determined by ROC analysis. The ROC curve showed that the optimal cutoff values were 2.18 for NLR (area under the curve (AUC), 0.653; 95% confidence interval (CI), 0.508–0.799; *p* = 0.0554, with sensitivity of 78.6% and specificity of 54.3%); 153.7 for PLR (AUC, 0.547; 95% CI, 0.362–0.731; *p* = 0.5597, with sensitivity of 50.0% and specificity of 61.8%); 0.025 for CAR (AUC, 0.714; 95% CI, 0.603–0.825; *p* = 0.0074, with sensitivity of 71.4% and specificity of 60.8%); and 48.4 for PNI (AUC, 0.644; 95% CI, 0.472–0.816; *p* = 0.0724, with sensitivity of 64.3% and specificity of 66.3%). Patients were subsequently divided into two groups based on the optimal cutoff values: a high group ≥ the optimal cutoff values, and a low group < the optimal cutoff values.

### 3.3. Correlation of Clinical Oncological Parameters with Inflammatory Biomarkers and Nutrition Status

The association between the baseline clinicopathological characteristics and NLR, PLR, CAR, PNI, and GNRI is summarized in Table 1. NLR was significantly associated with age, pathological T stage, and tumor grade. PLR significantly differed by gender and tumor grade. CAR was significantly associated with BMI, tumor size, pathological T stage, and tumor grade. PNI and GNRI significantly differed by age, gender, and tumor grade.

### 3.4. Survival Rates and Prognostic Factors

During the observation period, 27 patients developed metastatic recurrence, and 14 patients died, five of them from RCC. The results of Kaplan–Meier analysis for OS and MFS in all patients according to inflammatory biomarkers are shown in Figure 1. Significant difference in OS was observed between the low and high NLR groups (*p* = 0.0054, Figure 1a). However, no significant differences in OS were observed between the low and high PLR groups (*p* = 0.2624, Figure 1b) or the low and high CAR groups (*p* = 0.1308, Figure 1c). In contrast, significant differences in MFS were observed between the low and high NLR groups (*p* = 0.0094, Figure 1d), the low and high PLR groups (*p* = 0.0231, Figure 1e), and the low and high CAR groups (*p* = 0.0017, Figure 1f). The results of Kaplan–Meier analysis for OS and MFS in all patients according to nutrition indices are shown in Figure 2. There was a statistically significant difference in OS between the low and high PNI groups (*p* = 0.0310, Figure 2a), but not in MFS (*p* = 0.0509, Figure 2c). Additionally, the low GNRI group showed no significant difference in OS (*p* = 0.1138, Figure 2b), but significant difference in MFS (*p* = 0.0400, Figure 2d).

The results of the univariate analysis revealed that MFS correlated with NLR, PLR, CAR, and GNRI. The results of the multivariate analysis of the prognostic factors for MFS showed that only high CAR was an independent poor prognostic factor (hazard ratio (HR), 3.08; 95% CI, 1.24–7.67; *p* = 0.016) (Table 2). Although NLR and PNI were factors strongly associated with OS in univariate analysis, only high NLR was an independent poor prognostic factor for OS in multivariate analysis (HR, 3.96; 95% CI, 1.01–15.59; *p* = 0.049) (Table 3).

## 4. Discussion

Systemic inflammatory response and tumor microenvironment play an important role in the development and progression of cancer [5,19]. Accumulating evidence has shown that nutritional and immune status are involved in the development and progression of malignancies and thus influence survival outcomes [8].

The need for improved preoperative prognosis in localized RCC has remained significantly unmet. Promising blood markers, NLR and PLR, have been extensively researched in types of urological cancers. A recent meta-analysis indicated that an elevated NLR has been shown to be a significant predictor for worse OS (HR, 1.82; 95% CI, 1.70–1.94; *p* < 0.0001), progression-free survival (PFS) (HR, 1.89; 95% CI, 1.68–2.13; *p* < 0.0001), and recurrence-free survival (RFS) (HR, 1.86; 95% CI, 1.65–2.10; *p* < 0.0001) in patients with RCC [20]. Additionally, higher PLR has been confirmed to be correlated with poor OS (HR, 1.01; 95% CI, 1.00–1.02; *p* = 0.010), cancer-specific survival (CSS) (HR, 1.21; 95% CI, 1.00–1.46; *p* = 0.05), PFS (HR, 1.44; 95% CI, 1.28–1.62; *p* < 0.00001), RFS (HR, 1.73; 95% CI, 1.11–2.71; *p* = 0.02), and disease-free survival (DFS) (HR, 1.63; 95% CI, 0.91–2.94; *p* = 0.01) in 15,193 patients with RCC according to another meta-analysis [21].

The role of systemic inflammation as a part of underlying mechanisms contributing to the progression of RCC is thought to include neutrophilia with relative lymphocytopenia or thrombocytosis [22]. Neutrophils can be actively involved in the process of tumor development, growth, and metastasis [23]. In addition, they can promote disease progression via direct action on tumor cells, or by stimulating angiogenesis [24,25]. Neutrophilia as an inflammatory response inhibits the immune system by suppressing the cytolytic activity of immune cells such as lymphocytes, activated T cells, and natural killer cells [26]. Lymphocytes are reflected in cell-mediated immunity and are essential for the anti-tumor immune response. Increased lymphocyte infiltration in tumor areas is associated with improved response to therapy and prognosis in patients with solid tumors [24]. Thus, lymphocytopenia may be a marker of a diminished antitumor response in the host. On the other hand, platelets also play a prominent role in cancer progression. Various growth factors secreted by platelets, such as vascular endothelial tumor growth factor, platelet-activating factor, and platelet-derived growth factor, further support tumor growth and metastasis [27]. 

Recently, high pretreatment CAR has been reported to be effectively predictive of worse survival in patients with RCC [7,28]. A recent meta-analysis including 2829 patients with RCC revealed that high pretreatment CAR was associated with worse OS (pooled HR, 2.14; 95% CI, 1.64–2.79; *p* < 0.001) and DFS/PFS (pooled HR, 1.75; 95% CI, 1.31–2.35; *p* < 0.001) [7]. Research has found that CRP produces inflammatory cytokines and chemokines that lead to cancer progression [29]. In particular, interleukin-6 increases the synthesis of acute phase proteins, including CRP, and decreases albumin production in the liver [30]. Serum albumin is an objective indicator of nutritional status and clinical inflammation, which means that its expression decreases under inflammatory conditions [31]. Since these two proteins are synthesized in hepatocytes, the combination of elevated acute-phase inflammatory protein and lowered chronic-phase inflammatory protein may be prognostically useful [7].

Combining lymphocyte count and serum albumin levels, PNI is considered to reflect both cancer-related malnutritional status and cancer-related immune status. As patients with low PNI may suffer from a weakened antitumor response, and therefore a lower survival rate, PNI helps clinicians predict the clinical outcomes in solid tumors [8,32]. A recent meta-analysis involving 7629 patients with RCC indicated that a decreased PNI was shown to be a significant predictor of worse OS (HR, 2.00; 95% CI, 1.64–2.42; *p* < 0.001), CSS (HR, 2.54; 95% CI, 1.61–4.00; *p* < 0.001), and DFS/PFS/RFS (HR, 2.12; 95% CI, 1.82–2.46; *p* < 0.001) [14]. Therefore, patients with low PNI should be managed with nutritional support and treated in a way that corrects their malnutritional status. In contrast, previous studies have highlighted the utility of the GNRI in assessing the physical health of elderly patients with chronic diseases, and GNRI can also reflect the nutritional status and systemic inflammation of elderly cancer patients [33]. Moreover, malnutrition is a common problem in cancer patients, and severe malnutrition can progress to cachexia, a type of malnutrition characterized by loss of lean body mass, muscle wasting, and decreased immune, physical, and mental function. Cancer cachexia remains a devastating syndrome affecting 50–80% of cancer patients and causing at least 20% of deaths [34]. A recent multicenter retrospective study including 4591 patients with RCC showed that preoperative low GNRI was an independent predictor of RFS and CSS [15].

This study comprehensively evaluated the prognostic biomarkers of NLR, PLR, CAR, PNI, and GNRI in surgically treated nonmetastatic RCC Japanese patients. In this study, multivariate analysis showed that an elevated CAR and high NLR were independent prognostic factors for poor MFS and worse OS, respectively. To the best of our knowledge, although the results have been inconsistent, this is the first study to comprehensively investigate five representative inflammatory and nutritional biomarkers in Japanese patients with nonmetastatic RCC. All of the biomarkers examined in this study can be easily obtained in whole blood tests and could serve as simple and cost-effective prognostic markers.

This study had several limitations. The optimal cutoff values for NLR, PLR, CAR, and PNI to predict survival differed from previous studies, and there is still no best method to determine the exact cutoff values. Moreover, no details of systemic treatment for patients with recurrent metastatic RCC after definitive treatment were included in this study because of the complicated sequential treatment using the various agents available presently. Additionally, recent significant advances in surgical treatment may have influenced these prognostic analyses. Finally, the sample size, number of events, and observation period may have been insufficient to determine precise statistical significance. 

However, this study confirmed that higher pretreatment NLR and CAR correlate with worse survival outcomes in patients surgically treated for nonmetastatic RCC. Further prospective large-scale studies are needed to confirm these results.

## 5. Conclusions

The results of this retrospective analysis showed that NLR is closely associated with survival and CAR with metastasis in Japanese patients with nonmetastatic RCC. Accordingly, the measurement of pretreatment blood-based inflammatory biomarkers may be useful for prognostic risk stratification of patients with nonmetastatic RCC undergoing surgical treatment.

## Figures and Tables

**Figure 1 biomedicines-11-00533-f001:**
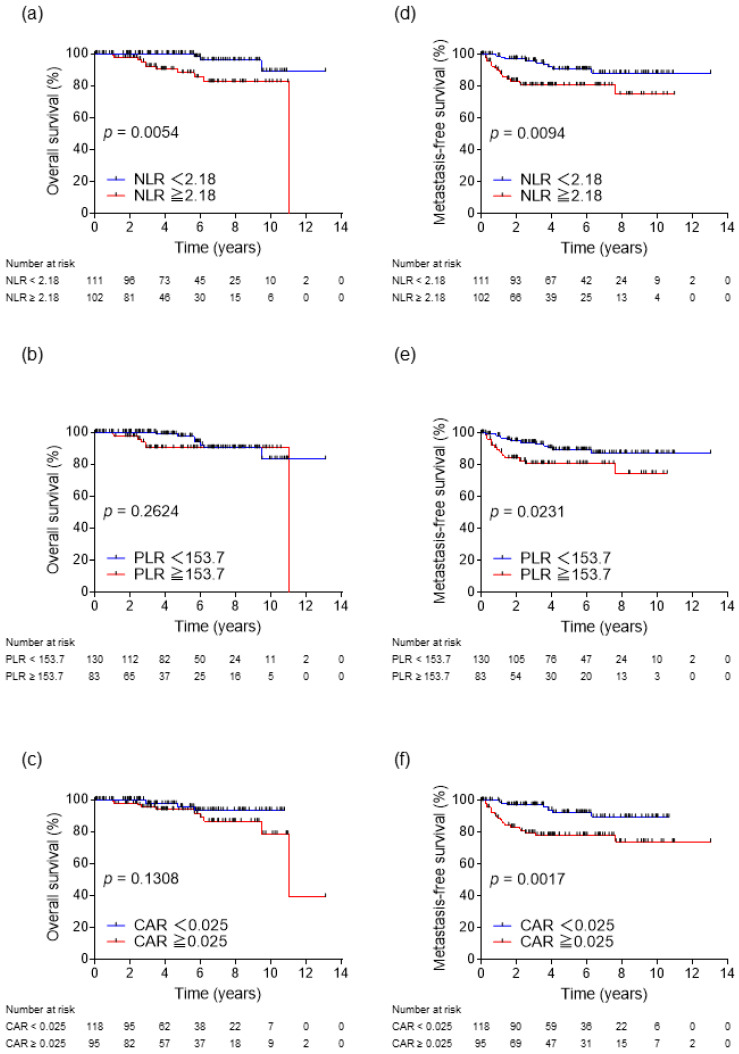
Kaplan–Meier analyses for overall survival (OS) and metastasis-free survival (MFS) based on NLR (**a**,**d**), PLR (**b**,**e**), and CAR (**c**,**f**). NLR, neutrophil-to-lymphocyte ratio; PLR, platelet-to-lymphocyte ratio; CAR, C-reactive protein-to-albumin ratio.

**Figure 2 biomedicines-11-00533-f002:**
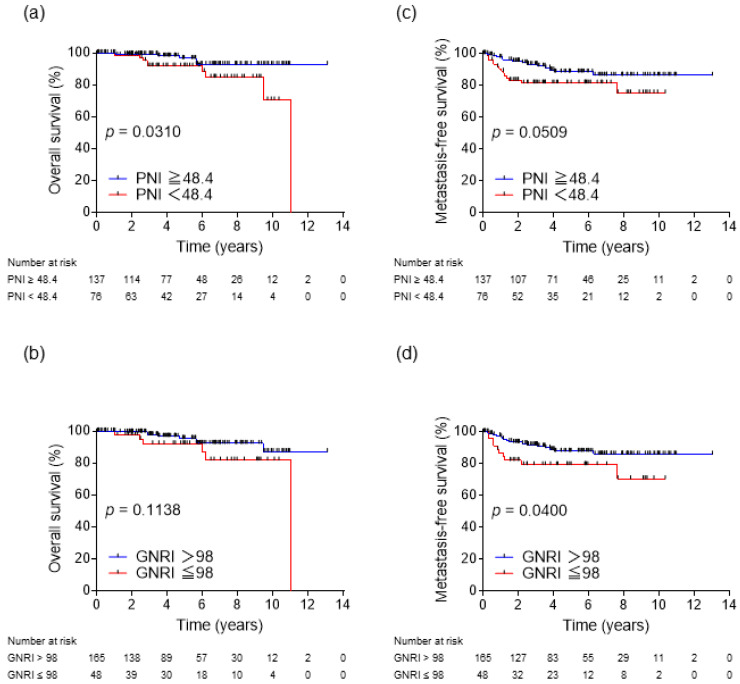
Kaplan–Meier analysis for OS and MFS based on PNI (**a**,**c**) and GNRI (**b**,**d**). PNI, prognostic nutritional index; GNRI, geriatric nutritional risk index.

**Table 1 biomedicines-11-00533-t001:** Association between baseline clinicopathological characteristics and NLR, PLR, CAR, PNI, and GNRI.

Variable	*n* (%)	NLR, Median (IQR)	*p*-Value	PLR, Median (IQR)	*p*-Value	CAR, Median (IQR)	*p*-Value	PNI, Median (IQR)	*p*-Value	GNRI, Median (IQR)	*p*-Value
Age			0.008		0.685		0.154		<0.001		<0.001
<65	113 (53.1)	1.97 (1.40–2.78)		139.2 (106.3–177.6)		0.023 (0–0.047)		52.0 (48.9–55.2)		105.7 (101.7–109.1)	
≥65	100 (46.9)	2.43 (1.72–3.21)		132.3 (105.6–179.9)		0.025 (0.022–0.050)		48.1 (44.0–51.6)		101.3 (96.5–105.7)	
Gender			0.484		0.001		0.231		0.036		0.026
Male	155 (72.8)	2.17 (1.64–2.96)		129.9 (98.3–172.5)		0.024 (0.021–0.049)		50.9 (46.9–54.6)		104.2 (99.8–108.3)	
Female	58 (27.2)	1.91 (1.36–3.20)		163.5 (121.7–214.3)		0.024 (0–0.044)		49.1 (44.6–53.0)		102.8 (95.3–106.4)	
BMI			0.681		0.213		<0.001		0.215		0.022
<25	139 (65.3)	2.05 (1.62–3.05)		140.0 (114.0–177.1)		0.023 (0–0.029)		50.5 (46.1–53.4)		102.8 (96.8–106.8)	
≥25	74 (34.7)	2.26 (1.63–2.96)		130.4 (92.6–182.6)		0.032 (0.022–0.070)		51.3 (46.5–55.0)		105.7 (100.9–108.7)	
Tumor size			0.393		0.557		<0.001		0.397		0.150
≤4 cm	157 (73.7)	2.05 (1.60–2.97)		136.0 (105.9–176.9)		0.023 (0–0.036)		50.8 (46.8–53.6)		104.2 (99.7–107.2)	
>4 cm	56 (26.3)	2.23 (1.69–3.34)		138.9 (106.1–214.2)		0.044 (0.023–0.095)		50.7 (43.6–54.1)		102.8 (93.9–108.7)	
Pathological T stage			0.015		0.126		<0.001		0.227		0.233
1–2	167 (78.4)	2.01 (1.55–2.88)		135.1 (100.9–176.7)		0.023 (0–0.044)		51.0 (46.7–54.0)		104.2 (99.6–108.0)	
3–4	32 (15.0)	2.56 (1.87–3.55)		140.8 (116.0–244.6)		0.064 (0.026–0.357)		50.2 (42.0–53.7)		100.9 (93.9–108.3)	
Unknown	14 (6.6)										
Histologic type			0.158		0.594		0.865		0.939		0.211
ccRCC	172 (80.8)	2.20 (1.61–3.06)		139.7 (107.2–179.2)		0.024 (0.021–0.047)		50.9 (46.4–54.0)		104.2 (99.7–108.0)	
Non-ccRCC	33 (15.5)	1.80 (1.63–2.55)		136.0 (96.4–183.3)		0.023 (0–0.100)		51.3 (45.9–53.4)		102.1 (95.5–107.2)	
Unknown	8 (3.8)										
Tumor grade			0.003		0.003		<0.001		0.001		0.007
<3	160 (75.1)	2.04 (1.56–2.88)		132.6 (101.0–174.8)		0.024 (0.005–0.043)		51.1 (47.1–54.2)		104.2 (99.8–108.0)	
≥3	34 (16.0)	2.58 (1.95–3.69)		170.4 (125.1–253.3)		0.050 (0.022–0.467)		47.0 (40.0–51.6)		101.0 (89.2–106.1)	
Unknown	19 (8.9)										
LVI			0.424		0.215		0.163		0.858		0.799
No	109 (51.2)	2.03 (1.56–2.83)		135.1 (98.0–172.1)		0.024 (0.021–0.036)		50.9 (46.0–54.6)		104.2 (98.3–108.3)	
Yes	89 (41.8)	2.26 (1.64–3.02)		141.0 (109.6–184.9)		0.025 (0–0.066)		51.0 (47.1–53.9)		104.2 (99.2–107.3)	
Unknown	15 (7.0)										

BMI, body mass index; CAR, C-reactive protein-to-albumin ratio; ccRCC, clear cell renal cell carcinoma; GNRI, geriatric nutritional risk index; IQR, interquartile range; LVI, lymphovascular invasion; NLR, neutrophil-to-lymphocyte ratio; PLR, platelet-to-lymphocyte ratio; PNI, prognostic nutritional index.

**Table 2 biomedicines-11-00533-t002:** Univariate and multivariate analyses of factors influencing MFS.

Variables		Univariate		Multivariate	
		*p*-Value	HR (95% CI)	*p*-Value	HR (95% CI)
Age	≥65 vs. <65	0.164	0.57 (0.25–1.26)		
Sex	Male vs. Female	0.105	2.41 (0.83–6.97)		
BMI	≥25 vs. <25	0.654	1.20 (0.55–2.61)		
Tumor size	>4 cm vs. ≤4 cm	<0.001	9.25 (3.91–21.90)		
Pathological T-stage	3–4 vs. 1–2	<0.001	9.84 (4.49–21.57)		
Histologic type	Non-ccRCC vs. ccRCC	0.611	1.27 (0.51–3.14)		
Grade	≥3 vs. <3	<0.001	10.39 (4.74–22.74)		
LVI	Yes vs. no	<0.001	9.01 (3.09–26.24)		
NLR	≥2.18 vs. <2.18	0.013	2.77 (1.24–6.17)	0.132	1.99 (0.81–4.87)
PLR	≥153.7 vs. <153.7	0.027	4.24 (1.95–9.24)	0.236	1.68 (0.71–3.95)
CAR	≥0.025 vs. <0.025	0.003	3.63 (1.54–8.59)	0.016	3.08 (1.24–7.67)
PNI	<48.4 vs. ≥48.4	0.056	2.09 (0.98–4.44)	0.605	0.75 (0.24–2.27)
GNRI	≤98 vs. >98	0.045	2.22 (1.02–4.85)	0.439	1.54 (0.52–4.56)

BMI, body mass index; CAR, C-reactive protein-to-albumin ratio; ccRCC, clear cell renal cell carcinoma; CI, confidence interval; GNRI, geriatric nutritional risk index; HR, hazard ratio; LVI, lymphovascular invasion; MFS, metastasis-free survival; NLR, neutrophil-to-lymphocyte ratio; PLR, platelet-to-lymphocyte ratio; PNI, prognostic nutritional index.

**Table 3 biomedicines-11-00533-t003:** Univariate and multivariate analyses of factors influencing OS.

Variables		Univariate		Multivariate	
		*p*-Value	HR (95% CI)	*p*-Value	HR (95% CI)
Age	≥65 vs. <65	0.930	1.05 (0.35–3.14)		
Sex	Male vs. Female	0.266	2.36 (0.52–10.68)		
BMI	≥25 vs. <25	0.812	1.14 (0.38–3.43)		
Tumor size	>4 cm vs. ≤4 cm	0.261	1.84 (0.63–5.36)		
Pathological T-stage	3–4 vs. 1–2	0.247	2.03 (0.61–6.71)		
Histologic type	Non-ccRCC vs. ccRCC	0.820	1.16 (0.31–4.31)		
Grade	≥3 vs. <3	0.598	1.43 (0.38–5.42)		
LVI	Yes vs. no	0.054	2.96 (0.98–8.89)		
NLR	≥2.18 vs. <2.18	0.012	5.11 (1.42–18.36)	0.049	3.96 (1.01–15.59)
PLR	≥153.7 vs. <153.7	0.268	1.81 (0.63–5.19)		
CAR	≥0.025 vs. <0.025	0.144	2.41 (0.74–7.82)		
PNI	<48.4 vs. ≥48.4	0.041	3.15 (1.05–9.43)	0.303	1.86 (0.57–6.06)
GNRI	≤98 vs. >98	0.124	2.32 (0.79–6.78)		

BMI, body mass index; CAR, C-reactive protein-to-albumin ratio; ccRCC, clear cell renal cell carcinoma; CI, confidence interval; GNRI, geriatric nutritional risk index; HR, hazard ratio; LVI, lymphovascular invasion; NLR, neutrophil-to-lymphocyte ratio; OS, overall survival; PLR, platelet-to-lymphocyte ratio; PNI, prognostic nutritional index.

## Data Availability

The data presented in this study are available on request from the corresponding author.

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
