# Peer review of "Comparison of the Prognostic Value of Inflammatory and Nutritional Indices in Nonmetastatic Renal Cell Carcinoma"

_biomedicines, 2023, doi:10.3390/biomedicines11020533_

Round 1

Reviewer 1 Report

The article shows the impact of pretreatment inflammatory and nutritional indices on the oncological outcomes in nonmetastatic renal cell carcinoma (RCC). The data is interesting albeit affected by many limitations. However, some minor modifications and improvements are required for the manuscript to be published.

Abstract:

Please start the abstract with a background.

Page 1, line 12: “Data were collected”. Harmonize this isolated sentence.

Page 1, lines 17-19: Furthermore, high NLR tended to be a worse prognostic 18 factor for overall survival (HR, 3.04; 95% CI, 0.89–10.37; p = 0.076). It must be clarified that the result is not statistically significant. Even in the text, quote the sentence clearly.

Introduction

Page 1, line 36: Delete “Comprehensively”.

Metodhs

Page 2, section 2.1: describe the criteria for exclusion and inclusion of patients.

Results

Page 5, line 118-121: “Although the low PNI group tended to show a worse prognosis for OS 119 (p = 0.080, Figure 2a)”. The result is not statistically significant, rephrase the sentence.

Page 7, line 135-136: “High NLR only tended to be a worse prognostic factor in 136 multivariate analyses (HR, 3.04; 95% CI, 0.89–10.37; p = 0.076) (Table 3)” The same for this sentence.

Discussion:

The lack of data on subsequent systemic therapies greatly limits the data on OS. Furthermore, adjuvant therapy is not mentioned. Is there any patient who received adjuvant therapy?

Based on the approval of pembrolizumab in the adjuvant setting, do you think the results could be altered by this systemic therapy? Are there any data in the literature? These aspects should be discussed.

Author Response

First of all, we would like to apologize for the significant revisions to the content. At the suggestion of the Reviewer, we calculated our own cutoff values of assessed indices and redid the statistical analysis based on them. Thus, text, figures, and tables were revised accordingly.

Abstract:

Please start the abstract with a background.

Page 1, line 12: “Data were collected”. Harmonize this isolated sentence.

Thank you very much for pointing out the mistake. The sentence was deleted.

Page 1, lines 17-19: Furthermore, high NLR tended to be a worse prognostic factor for overall survival (HR, 3.04; 95% CI, 0.89–10.37; p = 0.076). It must be clarified that the result is not statistically significant. Even in the text, quote the sentence clearly.

Thank you very much for insightful suggestions. We deleted the sentence of Page 1, lines 17-19, and corrected according to your suggestion (revised in page 1, line 22-23).

Introduction

Page 1, line 36: Delete “Comprehensively”.

Thank you very much for a comment. The sentence was deleted.

Metodhs

Page 2, section 2.1: describe the criteria for exclusion and inclusion of patients.

Thank you very much for an insightful comment. We described the criteria for exclusion and inclusion of patients according to your suggestion (revised in page 2, line 59-63).

Results

Page 5, line 118-121: “Although the low PNI group tended to show a worse prognosis for OS (p = 0.080, Figure 2a)”. The result is not statistically significant, rephrase the sentence.

Page 7, line 135-136: “High NLR only tended to be a worse prognostic factor in multivariate analyses (HR, 3.04; 95% CI, 0.89–10.37; p = 0.076) (Table 3)” The same for this sentence.

Thank you very much for important suggestions. According to your comments, the sentences were corrected, respectively (revised in page 6, line 157-158; page 10, line 179-182, respectively).

Discussion:

The lack of data on subsequent systemic therapies greatly limits the data on OS. Furthermore, adjuvant therapy is not mentioned. Is there any patient who received adjuvant therapy?

Based on the approval of pembrolizumab in the adjuvant setting, do you think the results could be altered by this systemic therapy? Are there any data in the literature? These aspects should be discussed.

Thank you very much for mentioning to important point. As you mentioned, there are no data regarding systemic therapy, and we need to be careful in interpreting the data on OS, which was also mentioned in “limitations” section. Furthermore, since no patients in this study received postoperative adjuvant therapies, the text has been added (revised in page 3, line 119-120).

This study is observational data prior to the approval of pembrolizumab as adjuvant therapy. Actually, of the 27 patients who developed metastatic relapse, 15 patients corresponded to the subjects of KEYNOTE-564 ClinicalTrials; 13 patients were classified as intermediate-high risk RCC, and 2 patients were classified as high risk. Therefore, if these patients had received pembrolizumab as adjuvant therapy, it might have affected OS and MFS.

Reviewer 2 Report

The study by Makino et al offers a valuable contribution in identifying new and perspective candidates for prognostic biomarkers in the field of RCC. However, there are certain issues that need to be addressed. In addition, minor grammar and spelling corrections should be performed.

Introduction Section

This section is quite brief. Can you please provide additional explanation regarding the role of inflammation and nutritional status as a part of underlying mechanism contributing to the progression of RCC?

Materials and Methods Section

The authors have decided to use the cut-off values of assessed indices stated in the literature. Please provide further explanation regarding the rationale behind such decision.

The authors should provide their own sample-specific cut-off values based on the ROC performance measurement and AUC interpretation and mention them in the Discussions section of the manuscript and compare them with the cut-off values provided in the literature.

Results Section

Line 132. Please rephrase “predictor” to “prognostic factor”.

Line 152. Please rephrase “prediction” to “prognosis”.

Discussion Section

Lines 166. And 168 Please rephrase “predictive biomarkers” to “prognostic biomarkers” . The definitions can be found at the following link https://pubmed.ncbi.nlm.nih.gov/27010052/.

Author Response

First of all, we would like to apologize for the significant revisions to the content. At the suggestion of the Reviewer, we calculated our own cutoff values of assessed indices and redid the statistical analysis based on them. Thus, text, figures, and tables were revised accordingly.

Introduction Section

This section is quite brief. Can you please provide additional explanation regarding the role of inflammation and nutritional status as a part of underlying mechanism contributing to the progression of RCC?

Thank you very much for an insightful suggestion. In response to your suggestion, we have made significant revisions to the content (revised in page 1-2, line 38-48).

Materials and Methods Section

The authors have decided to use the cut-off values of assessed indices stated in the literature. Please provide further explanation regarding the rationale behind such decision.

Thank you very much for mentioning to important point. In this revised manuscript, we calculated our own cutoff values of assessed indices. The optimal cutoff values were determined using the point closest to (0,1) on the receiver operating curve (ROC) (revised in page 3, line 122-132).

The authors should provide their own sample-specific cut-off values based on the ROC performance measurement and AUC interpretation and mention them in the Discussions section of the manuscript and compare them with the cut-off values provided in the literature.

Thank you very much for an insightful comment. Using the ROC analysis, the optimal cutoff values were 2.18 for NLR, 153.7 for PLR, 0.025 for CAR, and 48.4 for PNI, respectively (revised in page 3, line 122-132). Accordingly, the Kaplan-Meier analysis and the Cox proportional hazards model were conducted again, and significant changes were made to the "Results" and "Discussion" based on the results of statistical analyses.

Results Section

Line 132. Please rephrase “predictor” to “prognostic factor”.

Line 152. Please rephrase “prediction” to “prognosis”.

Thank you very much for comments. These sentences were corrected (revised in page 10, line 177; revised in page 12, line 200, respectively).

Discussion Section

Lines 166. And 168 Please rephrase “predictive biomarkers” to “prognostic biomarkers”. The definitions can be found at the following link https://pubmed.ncbi.nlm.nih.gov/27010052/.

Thank you very much for educational comments. These sentences were corrected (revised in page 13, line 257, 260).

Round 2

Reviewer 2 Report

Dear Authors,

Thank you very much for making suggested changes.

Your work provides a valuable source for some review paper regarding this topic.

All the best!